# Evaluation of *Miscanthus × giganteus* Tolerance to Trace Element Stress: Field Experiment with Soils Possessing Gradient Cd, Pb, and Zn Concentrations

**DOI:** 10.3390/plants12071560

**Published:** 2023-04-05

**Authors:** Giulia Bastia, Karim Suhail Al Souki, Bertrand Pourrut

**Affiliations:** 1Environmental and Forestry Sciences, Department of Agriculture and Food Sciences, Alma Mater Studiorum—University of Bologna, Via Zamboni 33, 40126 Bologna, Italy; giulia.bastia2@studio.unibo.it; 2Department of Environmental Chemistry and Technology, Faculty of Environment, Jan Evangelista Purkyně University in Ústí nad Labem, Pasteurova 3632/15, 400 96 Ústí nad Labem, Czech Republic; karim.souki@ujep.cz; 3Laboratoire Écologie Fonctionnelle et Environnement (ECOLAB), Université de Toulouse, CNRS, INPT, UPS-ENSAT, Avenue de l’Agrobiopôle, F-31326 Castanet-Tolosan, France

**Keywords:** *Miscanthus × giganteus*, trace element contamination, phytoremediation, stress tolerance

## Abstract

*Miscanthus × giganteus* demonstrated good phytostabilization potentials by decreasing the trace elements (T.E.s) mobility and enhancing the degraded soil quality. Nevertheless, most of the published work was performed under controlled conditions in *ex situ* pot experiments and/or with soils being spiked. Hence, data about the plant’s tolerance to increased T.E. concentrations in real conditions is still scarce and requires further investigation. For this sake, a field experiment was established by cultivating miscanthus plants in three different agricultural plots representing gradient trace element (Cd, Pb and Zn) concentrations. Another uncontaminated plot was also introduced. Results showed that T.E. concentrations in the leaves were tolerable to the plant. In addition, no variations were detected between the miscanthus cultivated in the contaminated and uncontaminated soils at the level of antioxidant enzymatic activities (ascorbate peroxidase and superoxide dismutase), photosynthetic pigments (chlorophyll *a* and *b* and carotenoids), and secondary metabolites (phenolic compounds, flavonoids, anthocyanins, and tannins). These outcomes validate the high capacity of miscanthus to resist and tolerate contaminated conditions. Such results may contribute to further understanding of the miscanthus tolerance mechanisms.

## 1. Introduction

In the last century, with the development of economy and society, soil trace element contamination has become increasingly common worldwide. Anthropogenic activities are the main sources of excess T.E.s in soils. Consequently, several negative environmental impacts are generated on ecosystems and human health [1]. Remediating such polluted areas constitutes a great challenge due to the persistence and extension of pollution and the presence of multiple pollutants in sites [2]. Conventional physicochemical methods, such as excavation, soil washing, and removal of soil to landfills have proved to be inappropriate for cleaning large contaminated areas due to their high costs and negative impact on ecosystems [3]. Therefore, phytomanagement, which combines phytoremediation techniques and the economic valorization of the produced biomass crop, seems to be a promising strategy as it contributes to enhancing and/or restoring the ecological state of the contaminated area concurrently with reducing the risks associated with the pollutants [4]. The contaminated area in Northern France around the former lead smelter (Metaleurop Nord) is a vivid example of soils presenting multiple T.E. pollution precisely in the ploughed horizon (0–20 cm) [5]. These soils have been suffering for more than a century from the smelter’s atmospheric emissions (mainly Cd, Pb, and Zn) in which the degree of contamination detected in that area was 20–50 times higher than the regional background levels (0.4, 38.0, and 74.0 mg kg^−1^, corresponding to Cd, Pb, and Zn, respectively) [6]. As demonstrated by Nsanganwimana et al. [7,8], miscanthus species possess high phytostabilization potentials to restore ecosystems and maintain local farmers’ activities. In the past years, we have witnessed an increasing number of papers that shed light on the efficiency of miscanthus species in the context of phytostabilization of T.E. contaminated soils by enhancing agronomic and microbiological properties and decreasing the mobility of the elements [3,9,10]. However, few have focused on the plant’s reaction to the different types of pollutants; most have focused on T.E. contamination. For instance, Zgorelec et al. [11] established greenhouse pot experiments (18 kg of soil) under semi-controlled conditions in which soils were spiked with augmenting Cd and Hg concentrations. The experiment lasted for three growing seasons. Miscanthus growth parameters and biomass yield were studied in addition to the T.E. concentrations in the aboveground biomass. Their results showed certain negative impacts on the miscanthus parameters. Nurzhanova et al. [12] also set up an ex situ pot experiment (8 kg of soil) using both naturally contaminated and Zn/Pb spiked soils. Interestingly, the plant health monitoring was performed only in those planted in the spiked soils. The studied parameters included photosynthetic pigments (Chl *a* and *b* and carotenoids), intensity of transpiration, water content in the tissues, and T.E. concentrations in different plant organs and plant growth parameters. Results of this study also demonstrated that miscanthus health is negatively affected by the spiked T.E. Moreover, Korzeniowska and Stanislawska-Glubiak [13] launched a two-year microplot experiment (1 × 1 × 1 m) with soils artificially contaminated by elevated concentrations of Cu, Ni, and Zn for three consecutive years prior to miscanthus cultivation. The soils also received NPK fertilization. The plant yield and tolerance index were measured along with the bioaccumulation, biocanocentration, and translocation factors. Pogrzeba et al. [14] initiated a field experiment (0.25 ha) divided into subplots (4 × 4 m) with naturally contaminated Cd, Pb, and Zn soils. The authors aimed to study the T.E. accumulation in the aerial biomass after two years of cultivation without any stress biomarker investigation. Briefly, the vast majority of the publications were based on ex situ experiments led under controlled conditions with a limited quantity of soils in which the plant roots were limited to certain pot areas, and hence T.E. uptake could be increased and cause the overstressing of the soil/plant system [15]. Moreover, several experiments were also established in unnatural contamination cases by artificially spiking the soils, which did not reflect the real T.E. mobility in contaminated fields. Finally, most of the papers discuss the short-term effects of the pollution (not surpassing three growing seasons), in spite of the fact that the life span of the miscanthus plant exceeds 20 years. Therefore, different factors might influence the miscanthus’s health and reaction to T.E.-induced stress in pot experiments and/or artificially contaminated soils.

Taking into consideration the scarcity of data from fields with natural contamination, and assuming that the results could vary between real field contamination and overstressed plant/soil systems (pots and artificial contamination), the aim of the current work is to study and evaluate the mid-term impacts of T.E.-induced stress on *Miscanthus* × *giganteus* in naturally contaminated fields. For this aim, three naturally contaminated fields presenting gradient Cd, Pb, and Zn concentrations were selected along with an uncontaminated control site and cultivated by miscanthus plants. After more than five years of cultivation, the plant health was assessed through the measurement of a set of stress biomarkers, including antioxidant enzymatic activities (ascorbate peroxidase and superoxide dismutase) and photosynthetic pigments (chlorophyll *a* and *b* and carotenoids) together with certain secondary metabolites (phenolic compounds, flavonoids, anthocyanins, and tannins), in order to investigate the genuine plant-adaptive and growing capacities in natural field conditions to corroborate and deepen its great potential for phytostabilization.

## 2. Results

### 2.1. Soil Physicochemical and T.E. Concentrations

The soil physicochemical parameters, particle size distribution, and T.E. concentrations of the studied agricultural plots are presented in Table 1. As stated above, the degree of T.E. in the contaminated plots varied with their distance from the former smelter, and was 20 to 50 times higher than the regional background values (0.4, 38.0, and 74.0 mg kg^−1^ corresponding to Cd, Pb, and Zn, respectively). Contrarily, T.E. in MC soils met the regional background values. The lowest pH values were obtained in the MC soils (6.2 ± 0.1) and the highest were in M500 (7.7 ± 0.1). M500 soils contained the highest SOC content (35.7 ± 1.6 g kg^−1^) and were more clayey than the others. Nevertheless, silt particles were dominant in all of the soils ranging from 49.8% ± 9.0 in M200 to 69.8% ± 1.4 in MC. The highest nitrogen contents were obtained in M500 (2.2 ± 0.3 g kg^−1^), while M200 and M700 displayed the lowest concentrations (1.2 ± 0.2 and 1.1 ± 0.1 g kg^−1^, respectively). P_2_O_5_ concentrations ranged from 0.1 ± 0.0 g kg^−1^ in MC and M500 to 0.2 ± 0.0 g kg^−1^ in M200 and M700. Finally, the most elevated CEC and CaCO_3_ values were detected in M500 (32.8 ± 2.1 cmol^+^ kg^−1^ and 22.7 ± 2.8 g kg^−1^, respectively), whereas MC possessed the least (11.2 ± 1.2 cmol^+^ kg^−1^ and 1.1 ± 0.2 g kg^−1^, respectively).

### 2.2. Trace Element (T.E.) Leaf Concentrations

T.E. concentrations were measured for each plant sample (Table 2). Cd results were slightly above the limit of detection (0.4 mg kg^−1^) except for M700 that recorded around 50% augmentation. Lead values increased from 1.1 ± 0.1 mg kg^−1^ in MC plants to 12.85 ± 2.16 mg kg^−1^ in M700. Significant differences were detected only between the uncontaminated and contaminated samples. Analogous to the Pb results, Zn showed an increase linked to its concentration in soil, recording the highest value in M700 with increases of 42% between MC and M200, 25% between M200 and M500, and 16% between M500 and M700.

### 2.3. Antioxidant Enzymatic Activities

Figure 1 and Figure 2 show no significant differences in the APX and SOD activities between the plants cultivated in the control and contaminated soils. Values of the former ranged between 0.09 ± 0.01 and 0.18 ± 0.06 U mg^−1^ FW in M200 and MC, respectively. As for SOD, values fluctuated between 5.06 ± 0.90 and 9.15 ± 1.96 U mg^−1^ FW in M500 and M700, respectively.

### 2.4. Photosynthetic Pigments and Secondary Metabolites

As shown in Figure 3, chlorophyll *a* concentration in the leaves of the plants grown in the M200 plots (4.62 ± 0.23 mg g^−1^ FW) was significantly lower than the values in other plant leaves. With this exception, significant differences were not detected among the different treatments, in which values fluctuated between 6.23 ± 0.23 mg g^−1^ FW in MC and 7.34 ± 0.32 mg g^−1^ FW in M700.

Figure 4 shows that the highest chlorophyll *b* level (4.34 ± 0.20 mg g^−1^ FW) was recorded in the plants grown in the most contaminated plot (M700), with a significant 24% increase compared to the MC plants. No significant difference was found between the M500 and M700 plants or between the MC and M500 plants. On the other hand, statistical differences existed between the M200 plants (possessing the lowest values) and the other treatments, including a minimum of a 25% decrease in comparison to the MC plants.

Lowest carotenoid contents were obtained in M200 (0.11 ± 0.004 mg g^−1^ FW), with 30% decrease compared to M700 (Figure 5). Otherwise, no significant differences were detected between the treatments.

Phenolic compounds (Figure 6) results showed no significant differences between samples. Values ranged between 25.07 ± 1.13 and 32.47 ± 2.50 mg gallic acid g^−1^ FW in M200 and M700 plants, respectively.

Flavonoid content (Figure 7) showed no significant differences between MC and M200 (2228.18 ± 92.17 mg L^−1^ catechin g^−1^ FW), nor between MC and M500 (3448.89 ± 312.22 mg L^−1^ catechin g^−1^ FW). However, the highest value associated to M700, presented an increase by 39% compared to MC.

Despite the slight variations, anthocyanin content showed no significant differences between plant samples (Figure 8). Results fluctuated between 1.17 ± 0.19 mg cyanidin g^−1^ FW in M700 and 1.87 ± 0.27 mg cyanidin g^−1^ FW in M200.

Tannin concentration (Figure 9) showed a particular difference between M200 (90.97 ± 3.48 mg L^−1^ catechin g^−1^ FW) and M700 (116.21 ± 9.11 mg L^−1^ catechin g^−1^ FW) with a 22% augmentation. Otherwise, no significant variations were detected between treatments.

## 3. Discussion

Numerous papers have shed light on the ability of *Miscanthus × giganteus* to stabilize areas contaminated by trace elements by primarily accumulating them in the underground parts of the plant, decreasing the bioaccessibility in the aerial parts and contributing to the restoration of the soil’s agronomic parameters [3,11]. These outcomes were mainly demonstrated as results of pot experiment scenarios, either with real contaminated or spiked soils.

Particularly in the context of the Metaleurop Nord contaminated site, many articles focusing on the phytoremediation potentials (based on in situ and ex situ experiments) of *Miscanthus × giganteus* have been published. These papers investigated the impacts of the plant on enhancing the soil quality, as well as the trace element behavior (mobility, speciation, and distribution in different plant organs) [7,8,16,17,18]. Taking into consideration that the adaptation to such stressful conditions can cause strong physiological, enzymatic, and molecular alterations, it was essential to verify whether *Miscanthus × giganteus* undergoes similar modifications under real-case contaminated field conditions.

To begin with, upon confirming the fact that miscanthus plants accumulate the highest portion of T.E.s in their roots [16], and due to choosing plant leaves to assess the miscanthus health, the option of studying T.E. concentrations exclusively in the leaves was chosen in the current work. As reported in Table 2, the T.E. concentrations increased in a dose-dependent manner. These results corroborate what has been previously observed in other in situ and ex situ experiments on several contaminated plots with gradient T.E. concentration surrounding the Metaleurop site [8,15,19]. However, the leaf T.E. concentrations observed in the pot experiment of Al Souki et al. [15,19], who used the same soils from the experimental fields as well as the same miscanthus cultivar, were over two times higher than the obtained results. The increase of T.E. concentrations in the leaves could be attributed to the similarities with the nutrient cations which lead to their absorption at the root surface and thus their uptake and translocation. In addition, other factors could also increase T.E. mobility and accessibility, such as the proton secretion by the roots which further acidifies the soil, in addition to the released root organic acids which enhance the T.E. solubility and thus availability [20]. Nevertheless, it is well known that miscanthus accumulate the highest T.E. concentrations in the rhizosphere. Hence, as demonstrated previously, these augmentations did not represent a critical threat to the plant’s health.

The extensive exposure of plants to elevated doses of T.E. leads to the disruption of the balance between the reactive oxygen species (ROS) and the antioxidant defense systems. Consequently, ROS are accumulated in the plants and oxidative stress is triggered [21]. ROS include ionic free radicals (superoxide anion, O_2_^−^; hydroperoxyl radical, HO_2_●; alkoxy radical, RO●; hydroxyl radical, ●OH) and nonradical molecules (hydrogen peroxide, H_2_O_2_; singlet oxygen, ^1^O_2_), which are strong oxidative agents that might cause irreparable damage to biomolecules such as DNA, proteins, cell walls, hormones, and lipids, thereby affecting plant growth and development, and eventually leading to programmed cell death [22,23]. Simultaneously, plants are forced to adapt and cope with over-generated ROS to sustain the cellular redox homeostasis via multiple ROS scavenging mechanisms, comprising enzymatic (superoxide dismutase, SOD; catalase, CAT; ascorbate peroxidase, APX; glutathione peroxidase, GPX), and non-enzymatic (ascorbic acid, AsA; glutathione, GSH; phenolic compounds, carotenoids, nonprotein amino acids) antioxidant defense systems [24].

The choice of plant health/stress indicators in the current work was related to an ongoing project in the laboratory that aimed to develop a high-throughput multiparameter set of plant-stress biomarkers that could determine different parameters within the shortest time frame and with the least costs possible. At that time, the parameters presented in the paper (that cover both the enzymatic and non-enzymatic antioxidant pathways) were developed and validated.

To begin with, the SOD (rapidly converting ^·^OH to H_2_O_2_) and APX (converting H_2_O_2_ to water and dioxygen) [21] results illustrated the absence of significant differences between the miscanthus species cultivated in contaminated and uncontaminated soils. This outcome contradicts the ones obtained by Zhang et al. [25] who recorded a 1.82-fold increase in SOD activities in the leaves of miscanthus grown in Cd-spiked soils compared to the uncontaminated controls. The SOD and APX results also oppose the ones obtained in the pot experiment of Al Souki et al. [15,19], who obtained a high increase in antioxidant enzymatic activities, especially in the most contaminated soils.

Different groups of phenolic compounds are present in plants. Among them are the simple phenolic compounds, tannins, flavonoids, and anthocyanins. It is noteworthy to mention that the phenolic compounds are efficiently involved in the plant’s non-enzymatic defense mechanism due to the hydroxyl and carboxyl groups that contribute to T.E.’s chelation [26]. Tannins also exhibit detoxification features by forming variable affinity complexes with T.E.s [27]. Moreover, flavonoids’ essential role is to promote the lower production of ROS [28]. Anthocyanins can also be produced as a result of excess T.E. levels and boost the plant’s antioxidant response to sustain their regular physiological state [29]. Generally, the secondary metabolites exhibited a similar approach to the antioxidant enzymes, with a few exceptions in flavonoids and tannins, which were more elevated in M700 plants. A different outcome was observed in the pot experiment of Al Souki et al. [15], in which the secondary metabolites increased in a dose-dependent manner (recording more than two times the augmentation in M700 compared to MC) after two years of cultivation.

Plant photosynthetic pigments are frequently considered when assessing the plant health upon T.E. exposure [20]. Carotenoids are also among the nonenzymatic antioxidants as they are involved in protecting chlorophyll pigments and plant organs through dissipating excess excitation energy, such as heat or scavenging ROS, and restraining lipid peroxidation [29]. In the current work, the photosynthetic pigments witnessed more explicit variations. Nonetheless, these significant variations were in the plants of M200 plots. No differences were detected between the control plants and those cultivated in the most contaminated plots. These results contrast with the outcome obtained in the T.E. hydroponic and spiked soil experiments of Guo et al. [30] and Zhang et al. [25], in which the contents of chl *a* and *b* and carotenoids were the lowest at the highest T.E. concentrations. Al Souki et al. [15] also recorded an average of a 58.3% drop in the most contaminated pots after two years of cultivation.

With a holistic approach to the obtained stress biomarker results, it is utterly evident that *Miscanthus × giganteus* demonstrated no signs of oxidative stress. The scarce exceptions were the variations existing between the plants in the uncontaminated MC plot and those in the most contaminated M700 plot, where the concentrations of chlorophyll *b*, flavonoids, and tannin contents increased by 24%, 39%, and 17%, respectively. Indeed, this might be a positive sign of adaptation and activation of the miscanthus defense mechanisms against T.E. stress. Thus, miscanthus plants possess high resistance and adaptation potentials in real-case T.E. contamination scenarios. The main reason behind the difference in the intensity of response could be attributed to the T.E. accumulation in the plant leaves. As stated before, the same miscanthus cultivar accumulated twice as much T.E. in the leaves when grown in pots (with a limited volume and soil quantity) than in the field. A similar trend could be observed in spiked soils and hydroponic solutions. The differences in T.E. uptake between field and controlled pot conditions may be linked to the different physiological state of the plant and/or to some modifications of soil parameters in the pot conditions. To begin with, the soils used in the pots should be dried before launching the experiment. Soil drying modifies the distribution of T.E. fractions, resulting in an increase in the water-soluble and exchangeable fractions [31,32]. Moreover, soils should be sieved and coarse fragments broken up. The homogeneous mixture resulting from this preparation may increase the contact between the roots, soil, water, and the soil matrix [33].

On the other hand, miscanthus plants in the field were older (more than five years of cultivation) with a higher biomass than the ones in the pot experiment (not exceeding three growing seasons). Typically, plants may be more vulnerable to T.E. in the early stages of their life cycle [34]. Lower T.E. concentrations in older plants can also be explained by the phytodilution process due to the higher biomass present in the field than in the pots [35]. In addition, the differences in root exudates or other rhizosphere processes may contribute to the lower accumulation in the field. Root exudates can modify the soil parameters around the roots [32]. In addition, adult plants in the field do not only have a more developed root system and more time to modify the rhizosphere, but can also invest more into exudation than the younger plants [32]. Finally, in the field, miscanthus roots can progressively extend downwards into the deeper soil layer (more than 2 m depth) which is less contaminated, resulting in a decrease in T.E. uptake with time, unlike the situation in the pots [36], which could also justify the absence of stress in the misanthus plants cultivated in the contaminated fields. This fact was confirmed by Sterckeman et al. [37], who demonstrated that the soil contamination in the agricultural plots in the Metaleurop site was limited to the ploughed horizon (0–20 cm).

## 4. Materials and Methods

### 4.1. Experimental Plots Characterization

Four agricultural plots representing gradient T.E. concentrations were cultivated by *Miscanthus × giganteus* with a density of 20,000 plants per ha^−1^. Rhizomes were obtained from Bical France (currently NovaBiom). The plots were designated as MC, M200, M500, and M700, respectively, according to their approximate Pb concentrations [5]. MC (1.3 ha) is the uncontaminated reference plot located in Linzeux village (50°20′46″ N, 2°12′15″ E), 75 km away from the smelter. The other three contaminated plots were located in the vicinity of the former Pb smelter in Noyelles-Godault, Northern France. M200 (1.1 ha) and M500 (0.8 ha) plots were within 1.8 km southeast (50°24′52″ N, 3°01′51″ E, Courcelles-les-Lens) and 1.4 km northeast (50°25′49″ N, 3◦02′13″ E, Evin-Malmaison) from the former smelter, while M700 (0.7 ha) is the nearest with a distance not exceeding 1 km (50°26′15.0″ N, 3°01′5.7″ E) from the former contamination source. Extra information and figures about the experimental plots can be found in the Appendix A file.

To determine the soils’ physicochemical parameters, samples were collected from the ploughed horizon (0–20 cm) of the different experimental fields, and homogenized, dried, and sieved to pass through a 2 mm mesh prior to laboratory analysis. Particle size distribution was detected according to the French NFX 31-107 standard by sedimentation and sieving after organic matter destruction using H_2_O_2_. pH (H_2_O) was measured after mixing soil and deionized water (1:5, *v/v*) according to the ISO 10390 standard. Soil organic carbon (SOC) was extracted and measured according to the ISO 14235 standard. Total N content was determined by the dry combustion method according to ISO 13878. Available phosphorus (P_2_O_5_) concentration was measured according to the French NFX 31-161 standard and Joret and Hébert [38] through extraction in ammonium oxalate solution ((NH_4_)_2_C_2_O_4_, 0.1 M, pH = 7). Cationic exchange capacity (CEC) was determined after percolating CH_3_COONH_4_ (1 M, pH = 7) solution into soil samples followed by an extraction of the ammonium ions (NH_4_^+^) with sodium chloride (NaCl, 1 M) according to the NF X31-130 French standard. Total carbonate (CaCO_3_) content was determined by measuring the CO_2_ formed after HCl (4 M) addition according to the NF ISO 10693 standard.

Cd, Pb, and Zn pseudototal concentrations in the soils were determined through acid digestion by a digestion plate (HotBlock^TM^ Environmental Express, Charleston, SC, USA). Aqua regia solution (HCl + HNO3, 3:1, 6 mL) was added and the aliquot heated at 120 °C for 120 min. An internal reference sample and a certified soil reference (CRM 141, IRMM, Geel, Belgium) were added to secure the quality control of extraction and analysis. T.E. concentrations in the soil extracts were detected by atomic absorption spectrophotometry (AA-6800, Shimadzu, Kyoto, Japan).

### 4.2. Plant Sampling and Preparation

After eight years of plantation, in order to assess the potential miscanthus stress against T.E. contamination, three leaves were harvested from six random plants (forth, fifth, and sixth foliar stage) from each plot (MC, M200, M500, and M700) and immediately flash frozen into liquid nitrogen. Samples were then stored at −80 °C after arrival at the laboratory. To measure the T.E. concentrations, the other leaves were harvested, placed into plastic bags, and kept in a cool box. In the laboratory, leaves were washed three times with osmosed water to remove dust particles. They were then oven-dried at 40 °C for 48 h, and afterwards ground into fine powder using a knife mill (GM200, Retsch, Haan, Germany) before analysis.

### 4.3. Trace Elements Concentration Determination in Plant Leaves

T.E. concentration in leaves was determined according to Waterlot et al. [38], in which 300 mg of each sample were acid-digested with nitric acid (HNO_3_, 70%) and heated at 95 °C for 75 min. Following the addition of hydrogen peroxide (H_2_O_2_, 30%), the solution was heated for further 180 min prior to adding osmosed water. Cd, Pb, and Zn concentrations in the solutions were determined by atomic absorption spectrophotometry (AA 6800, Shimadzu, Kyoto, Japan). Quality control for chemical extraction and digestion was performed by including blanks and internal and certified (Polish Virginia tobacco leaves, INCTPVTL–6, Warsaw, Poland) reference materials.

### 4.4. Stress Biomarker Measurement (Miscanthus Health Parameters)

#### 4.4.1. Antioxidant Enzymatic Activities

The measurement and quantification of the antioxidant enzymatic activities were performed according to Al Souki et al. [39] by analyzing the enzymes spectrophotometrically using a plate reader (Thermo Scientific Multiskan™ GO, Illkirch-Graffenstaden, France). The protocol indicates manual punching of five foliar discs (0.5 cm diameter) per plant, collected from the frozen samples, and transferring them into 96 deep-well plates (2 mL) with one 4 mm diameter glass bead in each well, leaving three free wells to be taken as controls. Later on, the samples were ground twice under frozen conditions using a Mixer Mill MM 400 (Retsch, Haan, Germany) for 1.5 min at 30 Hz after the addition of 1 mL of ice-cold Tris extraction buffer pH 7.0 containing 0.01 M EDTA, 0.4 M PVP, 0.05 M ascorbate, 11.44 mM 2β-mercaptoethanol, and protease cocktail inhibitor. Subsequently, the samples were homogenized for 2 min at 15 Hz with the Mill MM 400 and then centrifuged for 15 min at 5000× *g* at 4 °C. After centrifugation, 10 μL of supernatant was collected to quantify the protein content while using the bovine serum albumin (BSA, Sigma, Saint-Quentin-Fallavier, France) as standard [40]. The total superoxide dismutase (SOD) activity was determined by measuring its ability to inhibit the photochemical reduction of nitro blue tetrazolium (NBT) according to the Giannopolitis and Ries method [41] and the ascorbate peroxidase (APX) activity was calculated by the decline of absorbance at 290 nm as a consequence of ascorbate oxidation [42].

#### 4.4.2. Photosynthetic Pigments and Secondary Metabolites

Photosynthetic pigments (chlorophyll *a* and *b* and carotenoids) and secondary metabolites (phenolic compounds, flavonoids, anthocyanins, and tannins) were also assessed spectrophotometrically according to Al Souki et al. [39]. Initially, two foliar discs (0.5 cm diameter) per plant from the frozen leaves’ tissues were collected using a manual punch and weighed. The foliar discs were placed into 96 deep-well plates (2 mL) with one 4 mm diameter glass bead, and then ground twice under frozen conditions using the Mixer Mill MM 400 for 1.5 min at 30 Hz. Upon adding 1.5 mL of ice-cold 95% methanol in each well, samples were homogenized for 2 min at 15 Hz with the MM400 grinder and incubated in the dark for 24 and 48 h at room temperature.

After 24 h of incubation, the samples were again homogenized for 2 min at 15 Hz with the MM400 and 100 μL were collected from the 2 mL plates and used for calculating the photosynthetic pigment (Chl *a* and *b* and carotenoids) concentrations upon measuring the solution’s absorbance at 470, 652, and 666 nm, respectively. Calculations were performed based on the extinction coefficients and equations reported by Lichtenthaler [43].

After 48 h of incubation, the plates were centrifuged at 5000× *g* for 5 min (time required to extract the secondary metabolism molecules).

The total phenolic compounds’ concentration was determined according to Folin Ciocalteu assay based on utilizing 200 μL of a reaction mixture containing 20 μL of supernatant, 50 μL of Folin Reagent (10% *v/v*), and 0.098 mM of Na_2_CO_3_. The mixture in the plates was incubated for 2 h in the dark at room temperature for color development. The absorbance was measured at 510 nm, and the concentrations were calculated based a standard curve of gallic acid.

Flavonoid content was verified through the aluminum chloride method using catechin as reference compound. The reaction solution hcontained 25 μL of methanolic extract, 0.00724 mM NaNO_2_, 0.01125 mM AlCl_3_ and 0.05 mM NaOH. After a mixture of 1 min, the plates were then read at 595 nm and the concentration was determined using a standard curve of catechin. The results were expressed as mg catechin equivalent (CE) per gram of leaf fresh weight.

Anthocyanins were measured using the differential pH method based on the fact that anthocyanin pigments change their color according to the pH. Two solutions with different pH values were prepared: one in potassium chloride buffer (0.2 M, pH 1.0) and another in sodium acetate buffer (0.4 M, pH 4.5). After 15 min of incubation in the dark at room temperature, the absorbance was read at 510 nm and 700 nm for all the plates. Results were expressed as mg of cyanidin 3-glucoside equivalent per gram of fresh leaf weight.

Finally, to determine the concentration of tannins, a solution was used containing 50 μL of methanolic extract and 100 μL of vanillin solution and the plates were kept in the dark for 15 min at room temperature. Then the absorbance was read at 500 nm. The concentration was calculated using a standard curve of catechin and the results were expressed as mg L^–1^ catechin equivalent (CE) per gram of fresh leaf weight.

### 4.5. Statistical Analysis

Analysis of variance (ANOVA) and multiple analysis of variance (MANOVA) were used to statistically analyze and compare the obtained data from the studied parameters (*p* ≤ 0.05). When significant differences existed, Tukey HSD test was used for pair-wise comparison. All the statistical analyses were performed using R and XLSTAT 2017 software.

## 5. Conclusions

After a series of papers studying the phytoremediation potentials miscanthus plants in ex situ or T.E.-spiked soil contexts, this in situ experiment investigated the effect of gradient T.E. (Pb, Cd, and Zn) concentration in soil on the stress induced in *Miscanthus × giganteus.* The plant demonstrated optimum adaptation and a good level of tolerance to T.E.-induced stress in in situ conditions without any significant increases in the antioxidant enzymatic activities, secondary metabolites, and photosynthetic pigments, except chlorophyll *b*, flavonoids, and tannins. These variations confirm that miscanthus has adaptive and stress-resistant capacities. In fact, it is important to highlight that the plant demonstrates a high capacity of adaptation in an experiment that was led directly in a contaminated area and in natural plant conditions. This signifies that the natural condition of the field in which miscanthus was established can bring benefits to the plant, despite the high level of contamination that the soil presents. Moreover, this work reflects the importance of in situ experiments to phytoremediation and it could be interesting and pivotal to further deepen this aspect with a comparison of in situ and ex situ results to validate or evaluate the differences between the two experiments.

## Figures and Tables

**Figure 1 plants-12-01560-f001:**
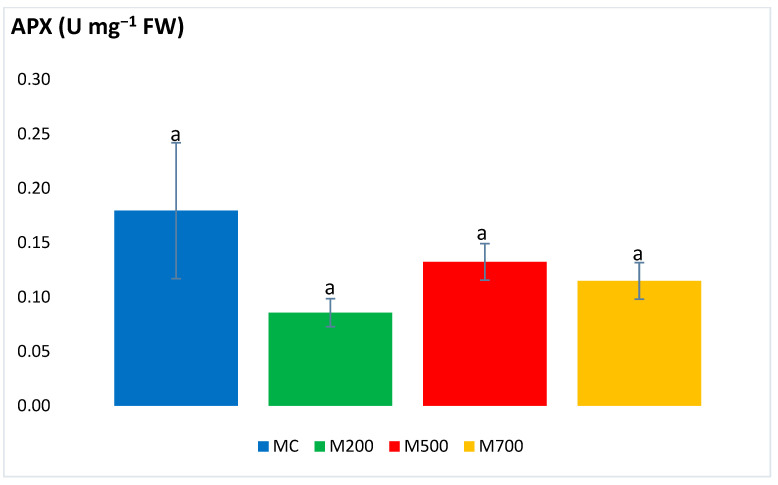
Ascorbate peroxidase (APX) activity in the leaves of Mis-B cultivar cultivated in soils with gradient T.E. concentrations (MC, M200, M500, M700). Values are presented as mean ± SE. Different letters refer to significant differences between plants (Tukey HSD test, *n* = 6, *p* ≤ 0.05).

**Figure 2 plants-12-01560-f002:**
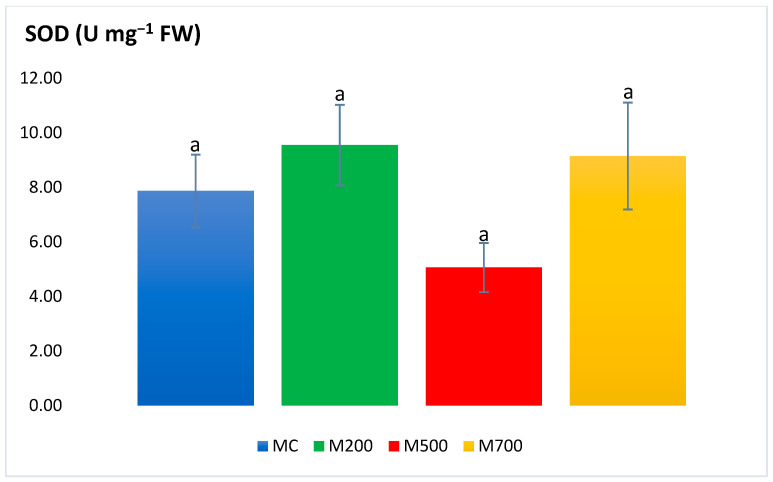
Superoxide dismutase (SOD) activity in the leaves of Mis-B cultivar cultivated in soils with gradient T.E. concentrations (MC, M200, M500, M700). Values are presented as mean ± SE. Different letters refer to significant differences between plants (Tukey HSD test, *n* = 6, *p* ≤ 0.05).

**Figure 3 plants-12-01560-f003:**
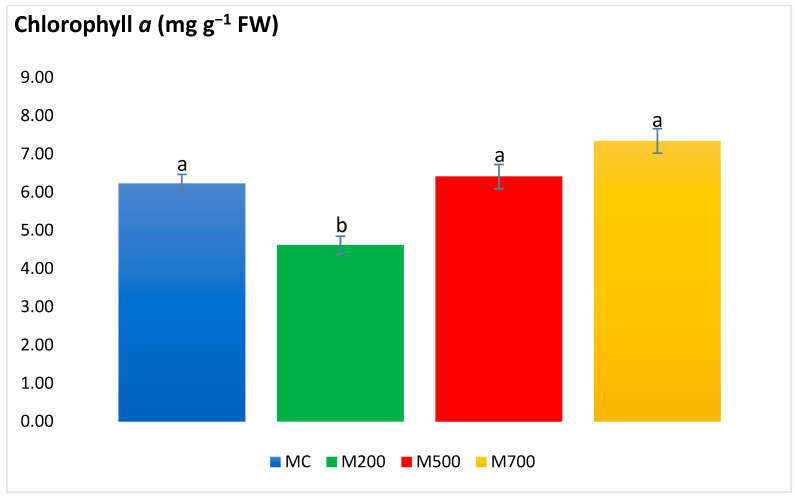
Concentration of chlorophyll *a* in the leaves of Mis-B cultivar cultivated in soils with gradient T.E. concentrations (MC, M200, M500, M700). Values are presented as mean ± SE. Different letters refer to significant differences between plants (Tukey HSD test, *n* = 6, *p* ≤ 0.05).

**Figure 4 plants-12-01560-f004:**
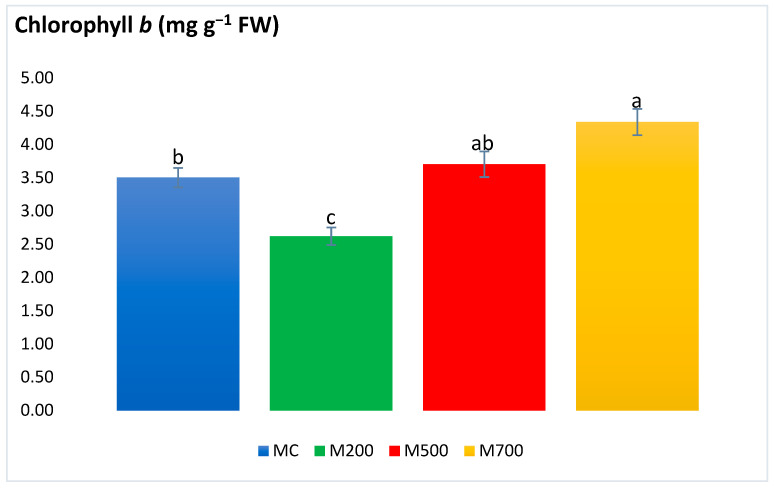
Concentration of chlorophyll *b* in the leaves of Mis-B cultivar cultivated in soils with gradient T.E. concentrations (MC, M200, M500, M700). Values are presented as mean ± SE. Different letters refer to significant differences between plants (Tukey HSD test, *n* = 6, *p* ≤ 0.05).

**Figure 5 plants-12-01560-f005:**
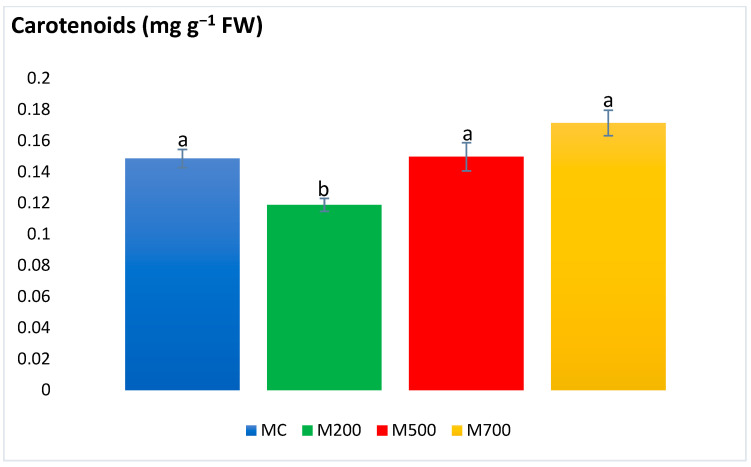
Concentration of carotenoids in the leaves of Mis-B cultivar cultivated in soils with gradient T.E. concentrations (MC, M200, M500, M700). Values are presented as mean ± SE. Different letters refer to significant differences between plants (Tukey HSD test, *n* = 6, *p* ≤ 0.05).

**Figure 6 plants-12-01560-f006:**
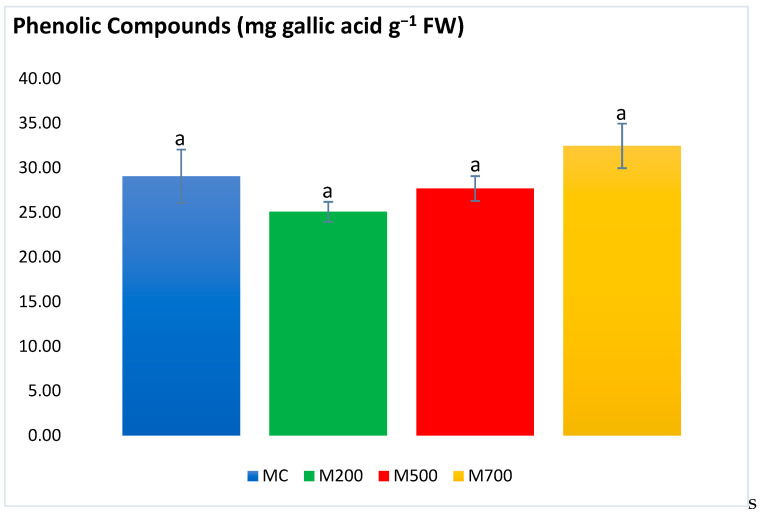
Concentration of phenolic compounds in the leaves of Mis-B cultivar cultivated in soils with gradient T.E. concentrations (MC, M200, M500, M700). Values are presented as mean ± SE. Different letters refer to significant differences between plants (Tukey HSD test, *n* = 6, *p* ≤ 0.05).

**Figure 7 plants-12-01560-f007:**
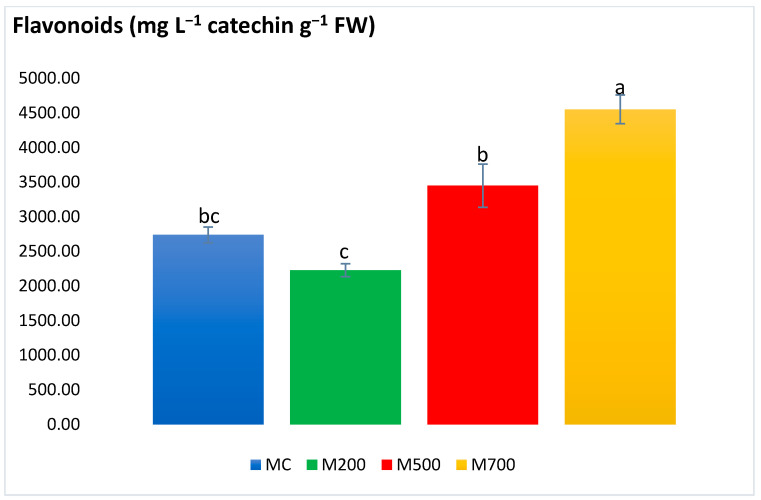
Concentration of flavonoids in the leaves of Mis-B cultivar cultivated in soils with gradient T.E. concentrations (MC, M200, M500, M700). Values are presented as mean ± SE. Different letters refer to significant differences between plants (Tukey HSD test, *n* = 6, *p* ≤ 0.05).

**Figure 8 plants-12-01560-f008:**
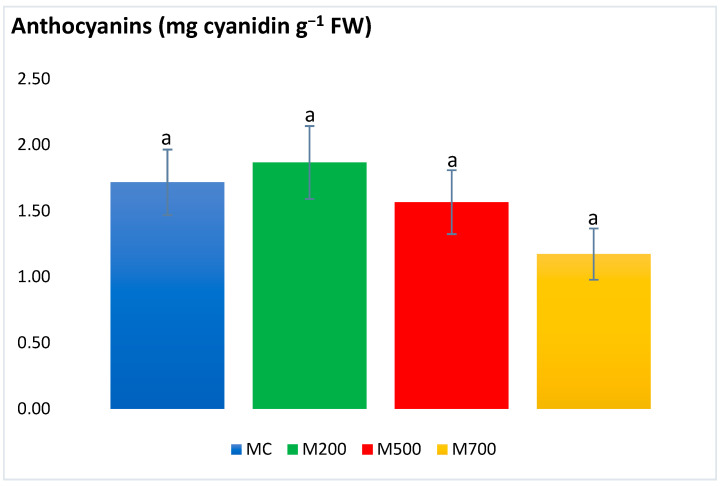
Concentration of anthocyanins in the leaves of Mis-B cultivar cultivated in soils with gradient T.E. concentrations (MC, M200, M500, M700). Values are presented as mean ± SE. Different letters refer to significant differences between plants (Tukey HSD test, *n* = 6, *p* ≤ 0.05).

**Figure 9 plants-12-01560-f009:**
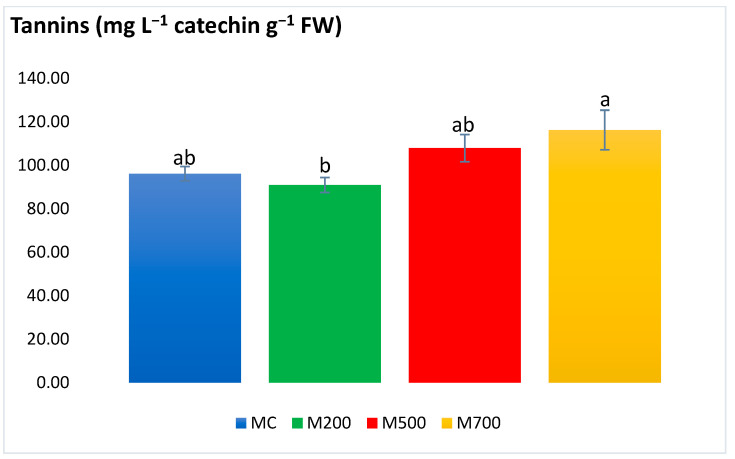
Concentration of tannins in the leaves of Mis-B cultivar cultivated in soils with gradient T.E. concentrations (MC, M200, M500, M700). Values are presented as mean ± SE. Different letters refer to significant differences between plants (Tukey HSD test, *n* = 6, *p* ≤ 0.05).

**Table 1 plants-12-01560-t001:** Soil physicochemical and trace elements (T.E.) concentrations in the studied plots (MC, M200, M500, M700). Results are presented as mean ± standard error (*n* = 6). SOC: soil organic carbon, TN: total nitrogen, P_2_O_5_: available phosphorus, CEC: cationic exchange capacity, CaCO_3_: calcium carbonates.

	MC	M200	M500	M700
Clay (%)	18.9 ± 1.1	21.3 ± 4.7	31.1 ± 1.9	19.5 ± 2.0
Silt (%)	69.8 ± 1.4	49.8 ± 9.0	52.2 ± 3.9	53.0 ± 2.5
Sand (%)	11.3 ± 0.9	28.9 ± 10.6	16.7 ± 5.2	27.4 ± 1.3
pH	6.2 ± 0.1	7.5 ± 0.5	7.7 ± 0.1	7.5 ± 0.4
SOC (g kg^−1^)	18.3 ± 3.7	16.4 ± 1.8	35.7 ± 1.6	18.2 ± 0.4
TN (g kg^−1^)	1.6 ± 0.1	1.2 ± 0.2	2.2 ± 0.3	1.1 ± 0.1
P_2_O_5_ (g kg^−1^)	0.1 ± 0.0	0.2 ± 0.0	0.1 ± 0.0	0.2 ± 0.0
CEC (cmol^+^ kg^−1^)	11.2 ± 1.2	16.3 ± 1.5	32.8 ± 2.1	14.9 ± 1.6
CaCO_3_ (g kg^−1^)	1.1 ± 0.2	7.7 ± 0.9	22.7 ± 2.8	10.2 ± 1.7
Cd (mg kg^−1^)	0.5 ± 0.0	5.1 ± 0.3	10.0 ± 0.8	15.0 ± 1.2
Pb (mg kg^−1^)	10.9 ± 0.8	203.5 ± 7.8	496.9 ± 12.7	703.8 ± 21.9
Zn (mg kg^−1^)	49.8 ± 3.9	329.0 ± 19.2	549.1 ± 14.2	1018.8 ± 27.3

**Table 2 plants-12-01560-t002:** Trace element (T.E.) concentrations in the leaves of Mis-B cultivar cultivated in soils with gradient T.E. concentrations (MC, M200, M500, M700). Values are presented as mean ± SE. Different letters refer to significant differences between plants (Tukey HSD test, *n* = 6, *p* ≤ 0.05).

	Cd (mg kg^−1^)	Pb (mg kg^−1^)	Zn (mg kg^−1^)
MC	0.42 ± 0.02 a	1.1 ± 0.1 a	31.41 ± 2.29 a
M200	0.43 ± 0.03 a	9.64 ± 2.29 b	44.73 ± 2.62 b
M500	0.46 ± 0.04 a	11.87 ± 1.48 b	55.80 ± 5.13 c
M700	0.69 ± 0.04 b	12.85 ± 2.16 b	64.60 ± 5.69 d

## Data Availability

Not applicable.

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
