# Peer review of "Evaluation of Miscanthus × giganteus Tolerance to Trace Element Stress: Field Experiment with Soils Possessing Gradient Cd, Pb, and Zn Concentrations"

_plants, 2023, doi:10.3390/plants12071560_

Round 1

Reviewer 1 Report (Previous Reviewer 1)

The authors have submitted a revised version of the manuscript with extensive text modification. However, I find data insufficient to prove a strong hypothesis. The experiment is quite interesting but, needs to incorporate more data, such as physiological performance, soil cation and cation exchange capacity etc. 

Author Response

Dear reviewer,

Thank you again for your time and given recommendations.

We would like to trigger your attention that the current work represents one of many articles focusing on the phytoremediation potentials of Miscanthus x giganteus in the context of the contaminated sites surrounding the former Pb smelter (Metaleurop Nord) in Northern France. Many papers (based on in situ and ex situ experiments) have been published studying the impacts of the plant on enhancing the soil quality (agronomic and biologic parameters) as well as the trace elements behavior (mobility, speciation). Upon fulfilling this phase, the second phase was studying the plant’s reaction against the T.E. stress, highlighted in the current paper. It is true that there could be more plant health parameters to be investigated (including the physiological performance), however, it is not possible to perform this analysis in the current time due to several constraints. An explanatory paragraph (highlighted) was added at the beginning of the discussion section in order to clarify and justify the purpose of not focusing on the different soil physicochemical and nutrient parameters.

As for the soil parameters, new data was added (CEC, CaCO3, TN and available P2O5). These data were provided by internal lab reports as part of the regular soil monitoring. Kindly refer to the modified text. New phrases in the materials and methods as well as results section are highlighted in yellow.

Examples of the papers focusing on the impacts of miscanthus on trace elements mobility and enhancing the contaminated soil quality in the context of the contaminated site in Northern France.

1) Potentials of Miscanthus x giganteus for phytostabilization of trace element-contaminated soils: Ex situ experiment (Nsanganwimana et al., 2021, https://doi.org/10.1016/j.ecoenv.2021.112125).

2) Metal accumulation and shoot yield of Miscanthus x giganteus growing in contaminated agricultural soils: Insights into agronomic practices (Nsanganwimana et al., 2015,  10.1016/j.agee.2015.07.023).

3) Metal, nutrient and biomass accumulation during the growing cycle of Miscanthus established on metal-contaminated soils (Nsanganwimana et al., 2015, https://doi.org/10.1002/jpln.201500163).

4) Interactions Sol-Plante dans un contexte de phytomanagement de sols pollués par des métaux : application à Miscanthus x giganteus (Karim Al Souki, thesis manuscript, https://www.theses.fr/2017LIL10041).

5) Assessment of Miscanthus x giganteus capacity to restore the functionality of metal-contaminated soils: Ex situ experiment (Al Souki et al., 2017, https://doi.org/10.1016/j.apsoil.2017.03.002).

6) Miscanthus x giganteus culture on soils highly contaminated by metals: Modelling leaf decomposition impact on metal mobility and bioavailability in the soil–plant system (Al Souki et al., 2020, https://doi.org/10.1016/j.ecoenv.2020.110654).

Reviewer 2 Report (Previous Reviewer 2)

The authors have addressed the comments and suggestions I made in the review. I have no further suggestions.

Author Response

Dear reviewer,

Thank you again for agreeing to review our paper and for your time and recommendations.

Reviewer 3 Report (New Reviewer)

Much work has been done, but the results are not so exciting.

In table 1, the four test soils are characterized, but nutrients P and N, mobile Mg and possibly Si are lacking; the latter may fix the metals within the cells walls of miscanthus, which is a Si accumulator. Also, the sampling depth of the soil is lacking. The metals come from aerial deposition, and are thus enriched at the surface layers. 8 years after plantation, however, roots may easily reach 2 m depth, according to your reference [10] - this is some kind of adaptation, and you should mention this in the section of discussion.

Differences in nutrient levels may overturn the metal effects. Thus, thus, at high Pb levels, Pb-phosphate gets precipitated in the soil. You can lower the effects of Pb by phosphate fertilizer addition.

The Pb concentration of leaves from the control soil is unusually high, in leaves from uncontaminated sites it is usually < 0,4 mg/kg. I think, this is a typing error, meaning 1,09 mg/kg. Cd in leaves is usually <0,2 mg/kg at uncontaminated areas. Check your analysis with a low-level reference sample! Zn data are plausible.

Author Response

Much work has been done, but the results are not so exciting.

Dear reviewer,

Thank you again for agreeing to review our paper and for your new comments and remarks that have enriched the text. We agree on most of the points, and we have modified the paper accordingly. However, we have certain justifications for the points that we do not agree on, hoping that we will convince you with our clarifications.

In table 1, the four test soils are characterized, but nutrients P and N, mobile Mg and possibly Si are lacking; the latter may fix the metals within the cells walls of miscanthus, which is a Si accumulator.

We would like to trigger your attention that the current work represents one of many articles focusing on the phytoremediation potentials of Miscanthus x giganteus in the context of the contaminated sites surrounding the former Pb smelter (Metaleurop Nord) in Northern France. Many papers (based on in situ and ex situ experiments) have been published studying the impacts of the plant on enhancing the soil quality (agronomic and biologic parameters) as well as the trace elements behavior (mobility, speciation). Upon fulfilling this phase, the second phase was studying the plant’s reaction against the T.E. stress, highlighted in the current paper. An explanatory paragraph (highlighted) was added at the beginning of the discussion section in order to clarify and justify the purpose of not focusing on the different soil physicochemical and nutrient parameters.

As for the soil parameters, new data was added (CEC, CaCO3, TN and available P2O5). These data were provided by internal lab reports as part of the regular soil monitoring. Kindly refer to the modified text. New phrases in the materials and methods as well as results section are highlighted in yellow.

Examples of the papers focusing on the impacts of miscanthus on trace elements mobility and enhancing the contaminated soil quality in the context of the contaminated site in Northern France.

1) Potentials of Miscanthus x giganteus for phytostabilization of trace element-contaminated soils: Ex situ experiment (Nsanganwimana et al., 2021, https://doi.org/10.1016/j.ecoenv.2021.112125).

2) Metal accumulation and shoot yield of Miscanthus x giganteus growing in contaminated agricultural soils: Insights into agronomic practices (Nsanganwimana et al., 2015,  10.1016/j.agee.2015.07.023).

3) Metal, nutrient and biomass accumulation during the growing cycle of Miscanthus established on metal-contaminated soils (Nsanganwimana et al., 2015, https://doi.org/10.1002/jpln.201500163).

4) Interactions Sol-Plante dans un contexte de phytomanagement de sols pollués par des métaux : application à Miscanthus x giganteus (Karim Al Souki, thesis manuscript, https://www.theses.fr/2017LIL10041).

5) Assessment of Miscanthus x giganteus capacity to restore the functionality of metal-contaminated soils: Ex situ experiment (Al Souki et al., 2017, https://doi.org/10.1016/j.apsoil.2017.03.002).

6) Miscanthus x giganteus culture on soils highly contaminated by metals: Modelling leaf decomposition impact on metal mobility and bioavailability in the soil–plant system (Al Souki et al., 2020, https://doi.org/10.1016/j.ecoenv.2020.110654).

Also, the sampling depth of the soil is lacking. The metals come from aerial deposition, and are thus enriched at the surface layers. 8 years after plantation, however, roots may easily reach 2 m depth, according to your reference [10] - this is some kind of adaptation, and you should mention this in the section of discussion.

Soil samples were collected from the contaminated ploughed horizon (0 – 20 cm). The phrase was added and highlighted in the Materials and methods section 2.1. It is true that the plant root reaching more than 2 m in depth is also another factor that could be taken into consideration while evaluating the plant reaction against the TE stress in the field. The phrase was added and highlighted in the last part of the discussion section.

Differences in nutrient levels may overturn the metal effects. Thus, thus, at high Pb levels, Pb-phosphate gets precipitated in the soil. You can lower the effects of Pb by phosphate fertilizer addition.

We agree with you that adding phosphate fertilizer can lower the Pb impacts. However, our objective was to assess the plant capacity to resist the TE stress without any adding any amendments. This is the reason behind not adding any kind of soil amendment. 

The Pb concentration of leaves from the control soil is unusually high, in leaves from uncontaminated sites it is usually < 0,4 mg/kg. I think, this is a typing error, meaning 1,09 mg/kg. Cd in leaves is usually <0,2 mg/kg at uncontaminated areas. Check your analysis with a low-level reference sample! Zn data are plausible.

Upon verification, we are very thankful, as the concentration of Pb was indeed 1.09 mg kg-1.

Reviewer 4 Report (New Reviewer)

Authors of the manuscript entitled, ‘plants-2274015’ showed the high capacity of Miscanthus ×Giganteus to resist and tolerate contaminated conditions. They concluded that their results contribute to understanding the miscanthus tolerance mechanisms to trace elements contaminated soils by decreasing their mobility. In my opinion, this research is novel. However, I also feel that there’s still a need for further revisions before the manuscript can be published.

Below are listed some of my comments.

1.  The whole document requires English revision, and there are still some grammatical errors and word omissions. Tense use is not consistent in some sections. Some sentences are too wordy and too long, causing them very hard to follow and understand; consider rephrasing.

2.  The abstract lacks a detailed summary of the findings; consider adding a substantial overview of obtained results. A statistical summary including numbers may be the best.

3. One of the mechanisms of plant tolerance to heavy metals is to limit the transport of heavy metals to the aboveground parts or cells. Why did the authors not consider measuring the heavy metal quantities in roots? In my opinion, knowing the root heavy metal quantities substantiates the leaf heavy metal quantities.

4. The result description can be enhanced, especially for Figures 3-9

Author Response

Authors of the manuscript entitled, ‘plants-2274015’ showed the high capacity of Miscanthus × Giganteus to resist and tolerate contaminated conditions. They concluded that their results contribute to understanding the miscanthus tolerance mechanisms to trace elements contaminated soils by decreasing their mobility. In my opinion, this research is novel. However, I also feel that there’s still a need for further revisions before the manuscript can be published.

Dear reviewer,

Thank you for your time and for accepting to review our work. Your remarks and recommendations that were taken into consideration while preparing the modified version of the manuscript.

Below are listed some of my comments.

  1. The whole document requires English revision, and there are still some grammatical errors and word omissions. Tense use is not consistent in some sections. Some sentences are too wordy and too long, causing them very hard to follow and understand; consider rephrasing.

English Language has been revised by Misses Layal El Wattar who holds a Bachelor degree in English Language and Literature (kindly find a copy of her certificate at the end of the letter). All of the modified sentences are in bold.

  1. The abstract lacks a detailed summary of the findings; consider adding a substantial overview of obtained results. A statistical summary including numbers may be the best.

In fact, we considered adding the numbers in the abstract since the beginning of writing the paper. However, due to the fact that 9 parameters besides the T.E. concentration in the leaves were used to assess the plant health measurement, it was a bit complicated to add digits. Nevertheless, the abstract was modified by adding the parameters name, so that the readers would get an idea about the work done by reading the abstract. Kindly find the modifications highlighted in the text.

  1. One of the mechanisms of plant tolerance to heavy metals is to limit the transport of heavy metals to the aboveground parts or cells. Why did the authors not consider measuring the heavy metal quantities in roots? In my opinion, knowing the root heavy metal quantities substantiates the leaf heavy metal quantities.

We would like to trigger your attention that the current work represents one of many articles focusing on the phytoremediation potentials of Miscanthus x giganteus in the context of the contaminated sites surrounding the former Pb smelter (Metaleurop Nord) in Northern France. Many papers (based on in situ and ex situ experiments) have been published studying the impacts of the plant on enhancing the soil quality (agronomic and biologic parameters) as well as the trace elements behavior (mobility, speciation and distribution in different plant organs). Upon fulfilling this phase, the second phase was studying the plant’s reaction against the T.E. stress, highlighted in the current paper. Upon confirming the fact that miscanthus plants accumulate the highest portion of trace elements in their roots in previous works, the choice in the current work was to study the concentrations in the leaves, as they were used to assess the plant stress/health parameters.

1) Potentials of Miscanthus x giganteus for phytostabilization of trace element-contaminated soils: Ex situ experiment (Nsanganwimana et al., 2021, https://doi.org/10.1016/j.ecoenv.2021.112125).

2) Metal accumulation and shoot yield of Miscanthus x giganteus growing in contaminated agricultural soils: Insights into agronomic practices (Nsanganwimana et al., 2015,  10.1016/j.agee.2015.07.023).

3) Metal, nutrient and biomass accumulation during the growing cycle of Miscanthus established on metal-contaminated soils (Nsanganwimana et al., 2015, https://doi.org/10.1002/jpln.201500163).

4) Interactions Sol-Plante dans un contexte de phytomanagement de sols pollués par des métaux : application à Miscanthus x giganteus (Karim Al Souki, thesis manuscript, https://www.theses.fr/2017LIL10041).

5) Assessment of Miscanthus x giganteus capacity to restore the functionality of metal-contaminated soils: Ex situ experiment (Al Souki et al., 2017, https://doi.org/10.1016/j.apsoil.2017.03.002).

6) Miscanthus x giganteus culture on soils highly contaminated by metals: Modelling leaf decomposition impact on metal mobility and bioavailability in the soil–plant system (Al Souki et al., 2020, https://doi.org/10.1016/j.ecoenv.2020.110654).

  1. The result description can be enhanced, especially for Figures 3-9

Phrases were amended in certain results according to recommendations. New phrases are highlighted in the modified version.

Round 2

Reviewer 1 Report (Previous Reviewer 1)

The authors have incorporated suggestions, accordingly.

Reviewer 4 Report (New Reviewer)

I am happy to see the revision of manuscript based on the comments. I recommend it to be accepted.

This manuscript is a resubmission of an earlier submission. The following is a list of the peer review reports and author responses from that submission.

Round 1

Reviewer 1 Report

The submitted manuscript entitled "Evaluation of the Miscanthus × giganteus tolerance to trace elements stress: Field experiment with soils possessing an increasing gradient concentration of Cd, Pb and Zn" contains preliminary data. The hypothesis of the experiment is weak. This type of several work has already been published. Why the authors have selected repetitive work? 

The manuscript entitled “Evaluation of the Miscanthus × giganteus tolerance to trace elements stress: Field experiment with soils possessing an increasing gradient concentration of Cd, Pb and Zn” is well written. However, in the manuscript it is difficult to find strong novelty. Also, the manuscript contains few data from some pathways and other from other stress inducing pathways. For example, the authors have only incorporated APX and SOD data, why? Reactive oxygen species data is missing in the manuscript. Additionally, below find a few articles, which are almost similar to the submitted manuscript.    

Nurzhanova, A., Pidlisnyuk, V., Abit, K., Nurzhanov, C., Kenessov, B., Stefanovska, T. and Erickson, L., 2019. Comparative assessment of using Miscanthus× giganteus for remediation of soils contaminated by heavy metals: a case of military and mining sites. Environmental Science and Pollution Research26, pp.13320-13333.

Zgorelec, Z., Bilandzija, N., Knez, K., Galic, M. and Zuzul, S., 2020. Cadmium and mercury phytostabilization from soil using Miscanthus× giganteus. Scientific Reports10(1), pp.1-10.

Korzeniowska, J. and Stanislawska-Glubiak, E., 2015. Phytoremediation potential of Miscanthus× giganteus and Spartina pectinata in soil contaminated with heavy metals. Environmental Science and Pollution Research22(15), pp.11648-11657.

Pogrzeba, M., Krzyżak, J. and Sas-Nowosielska, A., 2013. Environmental hazards related to Miscanthus x giganteus cultivation on heavy metal contaminated soil. In E3S Web of Conferences (Vol. 1, p. 29006). EDP Sciences.

Reviewer 2 Report

Specific comments:

1.      Were any soil properties other than metal content determined?

2.      Please provide the permissible contents of Cd, Pb, and Zn in soils. (Page 2)

3.      Please add references to all methods used.

4.      Please correct editing errors, e.g. table 2. leaf (…)

5.      Please provide the titles of the vertical axis of the figures (instead of placing them in the middle of the figures).

6.      Please move all the figures after their first citation.

7.      All abbreviations and acronyms used in tables and figures should be defined in the table notes or figure captions.

8.      Please use statistical terms instead of formulation “relatively higher” (Page 10), “slight exceptions” (Page 10).

Reviewer 3 Report

Row numbering is necessary, for further.

The Method section should provide full methodological information. 2.1. the subsection is a little bit chaotic – I suggest making a map of the study area. What was the size of the plots? What about agrotechnical treatments before planting and during cultivation (fertilization?). Additionally, the analysis of trace elements in the soil should be separately described (including the digestion process).

Table 1 should be supplied by other soil properties such as soil pH, and granulometric composition.

The photos of plants from each site are necessary and should be provided.

Fisher's post-hoc test is too weak and sensitive – I suggest using Tukey HSD.

The units - Flavonoids and Tannins (mg L-1 catechin g 1 FW) is not clear.

I am afraid that the presented research does not correspond with the current state of the art of the content and message brought =innovation. The method section needs considerable correction while the presentation of results and discussion is immature and weak.